# Gender policy and intimate partner violence in Colombia

**Dick Durevall**[ID]*

Department of Economics, School of Business, Economics and Law, University of Gothenburg, Gothenburg, Sweden

* dick.durevall@economics.gu.se

## Abstract

In 1995, Colombia signed the first legally binding international treaty that criminalizes all forms of violence against women. Subsequently, the government took several steps to improve laws and policies, but the progress was slow. This study uses a differences-in-differences approach and Demographic and Health Survey data to estimate the impact of a renewed effort to reduce intimate partner violence (IPV), based on recommendations by the UN. To identify the effect of the national policies, it uses the fact that while the central government passes laws and formulates policies, it partly relies on departments (provinces) to implement them. Of Colombia's 32 departments and Bogota D.C., approximately a quarter had some type of gender policy in place by 2011. The main finding is that self-reported intimate partner violence decreased from 20% to 16% between 2010 and 2015 in departments that had implemented IPV policies, while it remained at 19% in the others.

**Data Availability Statement:** Third party data was obtained for this study from The DHS Program. Data may be requested from The DHS Program after creating an account and submitting a concept note. More access information can be found on The DHS Program website (https://dhsprogram.

## Introduction

Intimate partner violence (IPV) is an entrenched and widespread issue on a global scale, with approximately one-third of women experiencing physical or sexual violence inflicted by their partners [1]. This form of violence not only causes direct harm to the physical and mental well-being of women but also carries far-reaching social and economic ramifications for families and society [2]. Furthermore, the impact of IPV extends beyond immediate victims, as children who witness such violence are more likely to find themselves in similar situations as adults, perpetuating a distressing generational cycle [3].

Despite this prevalence of IPV, the first legally binding international treaty that criminalized all forms of violence against women did not take place until 1994; this was called the Inter-American Convention on the Prevention, Punishment, and Eradication of Violence against Women (Convention of Belém do Pará). Since then, the UN General Assembly has, in several instances, called on the member states to adopt national gender policies that address IPV. In 2008, there was a renewed effort to do so when the UN launched the campaign"UNITE to End Violence against Women," as a multi-year effort aimed at preventing and eliminating violence against women and girls around the world [1, 2].

Most Latin American countries initiated a process of enacting laws and formulating gender policies as a result of the Convention of Belém do Pará [3, 4]. In Colombia, which has among

com/data/Access-Instructions.cfm). The authors confirm that interested researchers would be able to access these data in the same manner as the authors. The authors also confirm that they had no special access privileges that others would not have.

**Funding:** DD recived financial support from Vetenskapsrådet (the Swedish Research Council). The grant number is 201705684. The URL is https://www.vr.se/. The funder did not play any role in the study design, data collection and analysis, decision to publish, or preparation of the manuscript.

**Competing interests:** The author has declared that no competing interests exists.

the highest rates of IPV in Latin America and the Caribbean [5], this process was initially slow and uneven, and the commitments did not seem to significantly improve women's rights in general or reduce IPV [6–8]. However, following the UNITE campaign, the Colombian government took several steps to improve laws, policies, and reform institutions that manage gender policies. In particular, Colombia passed and amended laws against IPV in 2008 and 2011, which, according to Hoyos and Benjumea [9], marked a turning point. Furthermore, it launched national plans in 2010 and 2013, with several policy actions aimed at reducing IPV [6].

This study estimates the impact of Colombia's efforts to reduce IPV between 2010 and 2015 using data from Demographic and Health Surveys (DHSs) [10, 11]. In Colombia, the central government passes laws and formulates policies, while the departments (provinces) play an active role in implementing them [15, 16]. Thus, out of Colombia's 32 departments and the capital Bogota D.C., by 2010–2011, only eight departments had adopted a gender policy program, two had plans to adopt such a program, and another three departments had set up institutions that address gender issues more broadly [18]. This makes Colombia a suitable case for evaluating national gender policies using differences-in-differences (DiD), a quasi-experimental approach that compares changes in outcomes over time between a treatment group and a comparison group.

This study contributes to research by evaluating the initiatives of the government of Colombia and the UN to reduce IPV. To the best of my knowledge, this is the only study addressing this issue quantitatively in Colombia or other parts of the world. A number of studies have evaluated specific prevention programs in other countries, such as workshop activities, public-awareness campaigns, and cash transfers [25, 26], and some studies have evaluated the introduction of new laws, such as divorce laws [27–30], the criminalization of domestic violence [31], and female-oriented property rights [32]. Yet, it is widely recognized that there are many interconnected drivers of IPV and that prevention requires comprehensive strategies [33]. A combination of laws and a national plan with a multi-sectoral program, which engages with a range of stakeholders, as in Colombia, is therefore considered the most promising approach to preventing gender-based violence [34, 35]

My hypothesis is that the new and amended laws and national gender policies, implemented roughly between 2008 and 2013, combined with the departments' existing or newly adopted gender policies, affected IPV by 2015. The new laws were, according to Hoyos and Benjuméa [9], a turning point in the fight against IPV, and a large part of a long list of actions included in the national development plan, [6], and the national gender policy plan, were implemented [8].

## Background

The ratification of the Convention of Belém do Pará in 1995 required signees to establish mechanisms for the protection and defense of women's rights in terms of combating IPV. Consequently, the Colombian government introduced new gender policies, upgraded the agency working on gender issues, and enacted and amended laws related to IPV. However, many laws and policies were not enforced, and some put battered women in a weaker position than before [3, 7]. For example, after the year 2000, acts of domestic violence could only be prosecuted if the victim reported it to the authorities; the victim could stop the legal process, the accused might not go to prison even if they were convicted, and the crime was reconcilable, that is, the parties could reach an agreement instead of going to court. Thus, there was little progress in terms of preventing IPV in legal settings [6]. Moreover, there seems to be an agreement that other efforts to reduce IPV had a negligible effect [6–8, 18].

This lack of progress was not limited to Colombia; as a result, in 2008, the UN launched the campaign "UNiTE to End Violence against Women, 2008–2015," which was an initiative designed to spearhead an accelerated effort by the United Nations to address violence against women globally [2, 12]. This campaign aimed to persuade countries to adopt and enforce national legislation in line with international human rights standards, adopt and implement multi-sectoral national action plans, and conduct national and/or local awareness-raising campaigns. Moreover, the UN Entity for Gender Equality and the Empowerment of Women (UN Women) was established to, among other things, coordinate policies aimed at reducing violence against women [13].

The UN initiatives and efforts of civil society groups have contributed to several actions in Colombia. The government passed a set of comprehensive laws against IPV, most importantly in 2008 and 2011 [3, 9]. The new laws made it clear that it is the responsibility of the state to protect women's rights and establish legal procedures for victims. Domestic violence became a normal crime, as the legislation removed the possibility of reconciliation and the requirement that only the victim can file a report. A program managed jointly by the government and several UN agencies, Programa Integral contra la Violencia Basada en Género, was launched in 2008 [14]. In 2010, the government agency Consejería Presidencial para la Equidad de la Mujer was changed to Alta Consejería Presidencial para la Equidad de la Mujer, High Presidential Council for Women's Equity (ACPEM), and was granted the responsibility to assist the president and national government in designing policies that aim to improve gender equality [3, 6]. In the same year, the Colombian government launched a new national development plan, Plan Nacional de Desarrollo 2010–2014: Prosperidad para todos, which included a section on gender policy and domestic violence. And in 2013 Colombia launched a national plan for gender equality "Política de Equidad de Género para las Mujeres." A key objective of the national plans was to improve the implementation of gender policies, since it had become clear to the government that there was a gap between the policies and their actual implementation [6].

The national gender policy plan of Colombia covered the period 2013–2016 and consisted of six parts, where one part focused on reducing violence against women. This plan involved 17 government agencies and had 67 actions, such as the establishment of inter-institutional alliances and agreements with civil society, creation of institutions for monitoring and evaluation of policies and/or national plans, a program for collaboration with donors, and sensitization campaigns [6, 8].

One of the tasks of ACPEM was to support departments' development of gender policy programs [3], which can be seen as an integral part of national policies and a way to operationalize them. The formulation and adoption of departmental gender policy programs have proceeded slowly over time. Antioquia and Bogotá, D.C. adopted programs in 2002 and 2004, respectively, but Bogotá´s program was incomplete and replaced in 2010 [15, 16]; these were followed by Valle del Cauca and Nariño in 2008 and Tolima in 2009. In 2015, 14 departments had programs, and by 2017, the number had increased to 19. By 2011, ten departments had set up gender units of varying size and importance [17].

In Colombia, departmental gender programs closely follow national programs, which in turn are based on UN Resolutions (see, for example, [15]). A typical program consists of several policy actions that, among other things, aim to inform women about their rights, promote education programs that alter attitudes towards IPV, ensure enforcement of laws and policies, strengthen the capacity of institutions that investigate and prosecute IPV offenses, and improve access to shelters for battered women.

The national gender policy plan was evaluated with a focus on actions, that is, inputs, and about 75% of the actions had been conducted [8]. There is no information on the impact of

individual actions or the program, in general, on IPV. One reason for this might be paucity of good data on IPV in Colombia. Yet, the general view seems to be that IPV did not decline [18]. This is based on reports from a government agency that collects register data related to IPV (from hospitals and legal authorities). One report fails to uncover a trend in the number of reports of physical IPV given to the authorities throughout 2014–2016 [19], while another report finds a decline in physical IPV between 2009 and 2013 and then an increase in 2014 [20].

An alternative source of data is the DHSs. Gómez Lópes et al. [7] use DHSs from 2000, 2005 and 2010. They do not find that IPV had declined in Colombia, supporting the view of slow progress before the renewed effort. Yet, the share of women who reported their partner to the authorities increased from 11.8% in 2000 to 17.9% in 2005 and 19.3% in 2010; this casts doubt on the usefulness of detecting trends in data on reports about IPV that were given to the authorities. Bott et al. [5] report national prevalence rates of physical and sexual IPV using four Colombian DHSs from 2000, 2005, 2010, and 2015. According to their measures, the data show a steady but small decline in both physical and sexual IPV. For example, among ever-partnered Colombian women aged 15–49 years, their reported exposure to physical IPV declined from 40% in 2000 to 38.6% in 2005, 36.6% in 2010, and 32.3% in 2015. These conflicting results may be due to several factors, such as how the authors manage differences in the geographical coverage and how they specify their measures of IPV, as these vary to some extent across surveys.

## Materials and methods

### Data and outcomes

The main data sources for this study are DHS 2010 and DHS 2015. These are nationally representative, population-based, household surveys conducted using multistage clustered area sampling techniques. First, the country is stratified into major subnational regions from which census-based enumeration areas are selected. Then, households are randomly selected from the household list within each enumeration area. The response rates were 92.5% in DHS 2010 and 88.1% in DHS 2015, giving a total sample of women ever in a union of 81,340, of which 47,076 are from the DHS 2010 and 34,246 are from the DHS 2015. Detailed information about the survey design, sampling methods, and refusal rates are available in the DHS final reports [10, 11].

I use data from the DHS 2000 and DHS 2005, in addition to the DHS 2010 and DHS 2015, to evaluate the assumption of parallel trends, which must hold for the DiD approach to be valid. There is a DHS from 1995, but it does not have the same questions about exposure to IPV as the others. The DHS from 2000 also differs somewhat from the more recent ones: it covers 23 out of the 32 departments and only has information about whether a woman has ever experienced IPV but not about IPV during the past year.

Information on IPV was collected using a modified and abbreviated version of the Conflict Tactics Scale module. In every selected household, one randomly selected ever-married or ever-in-a-union woman aged 15–49, was asked about experiences with IPV from the current or last partner during the past year or at any point in time. Questions were asked only if privacy was guaranteed. More information about the Conflict Tactics Scale module is available in [21].

Exposure to IPV is measured by two binary variables: physical or/and sexual violence in the past year and physical or/and sexual physical violence ever. The binary variables are equal to one if the respondent answered yes to any of the questions. Physical and sexual violence are combined into one variable, since it is a commonly used measure. Moreover, very few women

were only exposed to sexual violence. The variable of main interest is IPV during the past year, since it measures exposure to violence after treatment. I also use IPV ever since it is available for a longer period and because it covers a larger part of the treatment period than IPV in the past year.

The actual questions asked are specific, which makes the results less culturally bound than general questions about violence. I use seven questions about physical violence: Did he push you or shake you? Did he hit you with his hand? Did he hit you with something that could hurt you? Did he kick or drag you? Did he try to choke or burn you purposely? Did he threaten or attack you with a knife, gun, or another weapon? Have you ever been attacked with a knife/gun or another weapon by your husband/partner? There is one question about sexual violence: Did he physically force you to have sexual intercourse with him when you did not want to?

## Method

To estimate the effect of laws and policies, I exploit the variation in gender policy across departments and Bogota, DC. Two departments with very small populations, San Andres y Providencia and Amazonas, were excluded because of a lack of data. I base the classification of departments on a review of regional gender policies in Colombia commissioned by the Spanish Agency for International Development Cooperation (AECID [17], one of the key aid agencies supporting Colombia's efforts to implement gender policy programs [3]. This study provides a detailed review of gender policy indicators in departments and municipalities up to 2011. Apart from adopted gender policy programs, it reports on the presence of an institutional setup for gender policies and a quota law, requiring that 30% of the members of the cabinet should be females. Institutional gender setups are ranked from 1 (an occurrence where there is one individual with little formal influence) to 5 (a proper gender office). S1 File describes the choice of treated and untreated departments in more detail.

Out of the 32 departments and Bogota, D.C., eight had a policy program in place, four had gender offices but no formal program, and two of these were in the process of implementing a gender policy program. All of these departments had at least 30% of women in the cabinet [17].

Whether a department has a gender policy depends on both the interest of the provincial government and whether support was provided by aid agencies. Thus, the adoption of a gender policy program or the presence of a gender office is not random. However, they are important components of the national government's efforts to reduce IPV. The key assumption is, therefore, that the classification of AECID [17] provides information on a key link between the national government and outcomes; this indicates the willingness and efficiency of the departments in implementing national gender policies.

Furthermore, I assume that there is a considerable time lag between the formal decision to adopt a gender policy program and its impact on IPV, which is likely to be a question of several years. Reviews of the implementation of gender policy programs support this assumption [16, 22]. Thus, it is unlikely that the adoption of a program in 2008 had an impact in 2010. Moreover, as mentioned, although previous efforts to reduce IPV may have had some effect, there seems to be general agreement that this was negligible [6–8, 18].

A related issue is the timing of the adoption of the gender policy. It is a challenge to pinpoint why some departments have implemented gender policies earlier than others, but in S2 File, I compare the characteristics of the departments in detail and show that several non-adopters have small populations. These departments, which are all in eastern and southern Colombia, are excluded from the main analysis. One likely reason for not focusing on gender policy is the small government and lack of resources, although there might also be other factors

as well. Thus, in the main specification. I have 12 treated and 11 untreated departments. As this amounts to only 23 departments (clusters), I report p-values calculated using a wild boot-strap procedure (the restricted version) when applicable [23], in addition to robust standard errors clustered at the department level.

The main DiD regression is specified as

$$IPV_{idt} = \beta_0 + \beta_1 Post_t + \beta_2(Post_t \; x \; Policy_d) + \beta_3 X_{idt} + \beta_4 Z_{dt} + \alpha_d + u_{idt} \qquad (1)$$

where (IPV) represents the outcome variable for woman $i$ in department $d$ included in the DHS survey $t$. $Post_t$ is an indicator that equals 1 for DHS 2015 and 0 otherwise; $Policy_d$ is 1 for all departments with a gender policy and 0 for the others. Individual and household control variables are represented by $X_{idt}$, variables at the department level are represented by $Z_{dt}$, and $\alpha_d$ is the department fixed effects. The coefficient of interest, $\beta_2$, shows the impact of gender policy on IPV. The linear probability model, that is, OLS, is used to estimate the regressions. The DHS sample design is accounted for using weights and clustering, as recommended [24].

The individual and household variables are age, ethnicity, educational attainment, house-hold wealth, household size, and rural or urban residence. These have been used in previous studies with the aim of controlling for potential differences in observable characteristics across women (see, for example [25–27]). The department fixed effects control for constant unob-served differences across departments, and survey-year fixed effects control for shocks and national reforms that impact all departments. The other department-level control variables are the log of real GDP per capita and an index of armed conflict. GDP per capita captures major changes in the economic environment that might affect IPV.

The index of armed conflict measures the average level of conflict between 2002 and 2013 and the decline in incidents that took place after that due to peace negotiations [28]. Several recent studies evaluate the impact of conflict on IPV and find that it increases IPV (e.g., [29–32]). Exposure to conflict in childhood or adolescence appears to increase the acceptance of violence in general and of IPV; women who are more exposed to conflict violence have more positive attitudes towards IPV and are more likely to stay in violent relationships. It may also affect the type of men available for marriage.

There are two studies on Colombia: Svallfors [33], who concludes that exposure to violence probably increased the acceptance of IPV, and Rieckmann [34], who argues that there was an increase in IPV due to an increase in the number of dysfunctional relationships. These long-run effects of armed conflict should be captured by department fixed effects, whereas the con-flict index should capture the impact of the sharp decline in armed conflict between 2010 and 2015.

Since it is possible there is social desirability bias, that is, some women in departments with gender policy recognize that IPV is illegal and are therefore either more or less willing to report it, I estimate the impact of gender policies on related outcomes: attitudes towards IPV among women (DHS 2010 does not have information about men's attitudes), whether couples in a union live together, if there are fewer planned divorces/separations, if IPV is less often than other reasons given as a reason for considering a divorce/separation, and if there is more help-seeking from formal institutions when mistreated. Moreover, since suspicion of unfaithfulness is closely linked to IPV [35], that is, it is probably the strongest predictor of IPV in the dataset, a change in the association between accusations of unfaithfulness and IPV could be an indica-tion of a behavioral change. Thus, I use a triple differences approach, interacting *Unfaithful-ness*, *Policy*, and *Post*, to evaluate how the relationship between accusations of unfaithfulness and IPV changed between 2010 and 2015.

To evaluate the assumption of parallel trends, I estimate event study models, use graphs and F-tests of linear trends for the pre-treatment period, and simulate the effects of deviations from the parallel trends assumption in the post-treatment period using the procedures developed by Rambachan and Roth [36].

To check the robustness of the results, I report the results when I include all departments (except the two with missing data), exclude departments without a formal gender policy program in 2011, and include all departments that had adopted a gender policy program by 2013. No department adopted a program in 2014 or 2015 [15, 17]. I also run 24 regressions, excluding one, and in one case, two, departments at the time. Finally, I restrict the sample to those in a union at the time of the survey. Unfortunately, the DHS 2010 does not have a male questionnaire [10], which limits the number of partner variables that can be used, but information about the partner's age and education is available. Although dropping women who did not have a partner at the time of the survey generates a selection bias since women are more likely to leave violent men than others, gender policies should also impact IPV among couples.

## Results

### Data description and summary statistics

Figs 1 and 2 provide descriptive statistics of the two key dependent variables, IPV past year and IPV ever, by treated departments (black) and untreated departments (stripes) in 2010. On average, there is a minor difference in prevalence between the two groups, 0.20 versus 0.19 for IPV in the past year, and 0.38 versus 0.34 for IPV ever. There is more variation within each group, but the differences between the departments are relatively small; the highest prevalence is about twice as high as the lowest.

Table 1 reports the descriptive statistics of the control variables by survey and by treated and untreated group of departments. The mean age is about 34 years in both surveys and groups. Moreover, the index of armed conflict is, on average, approximately the same in both

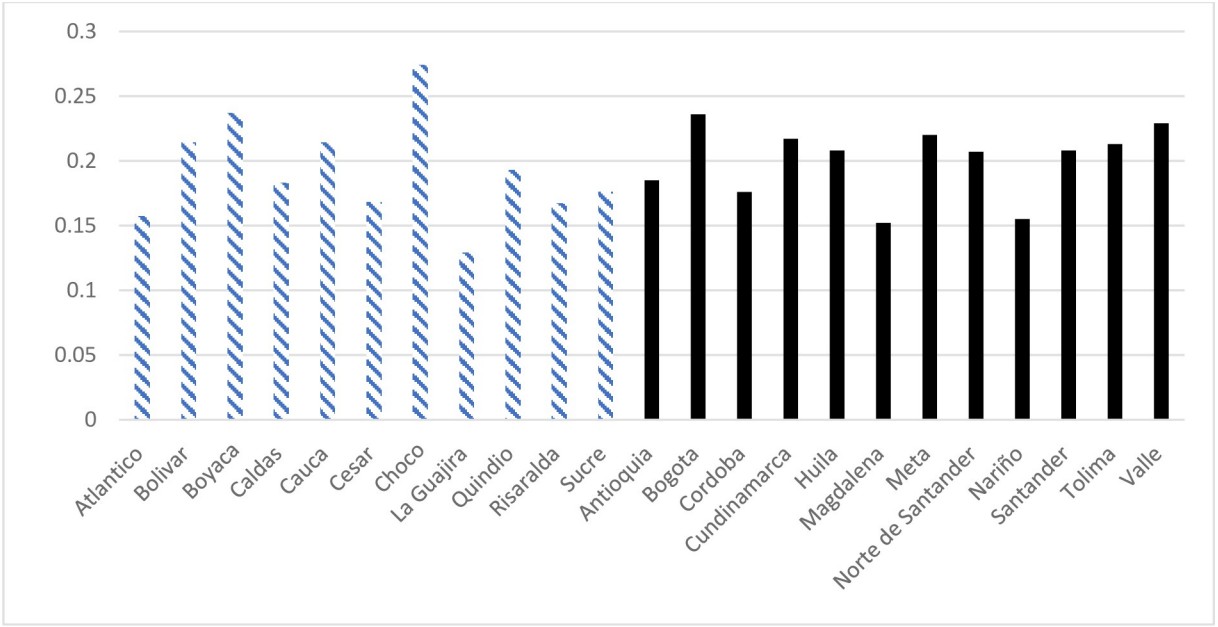

**Fig 1. Prevalence of self-reported IPV past year in 2010.** Treated departments in black and untreated departments in stripes. Note: Weights for domestic violence used.

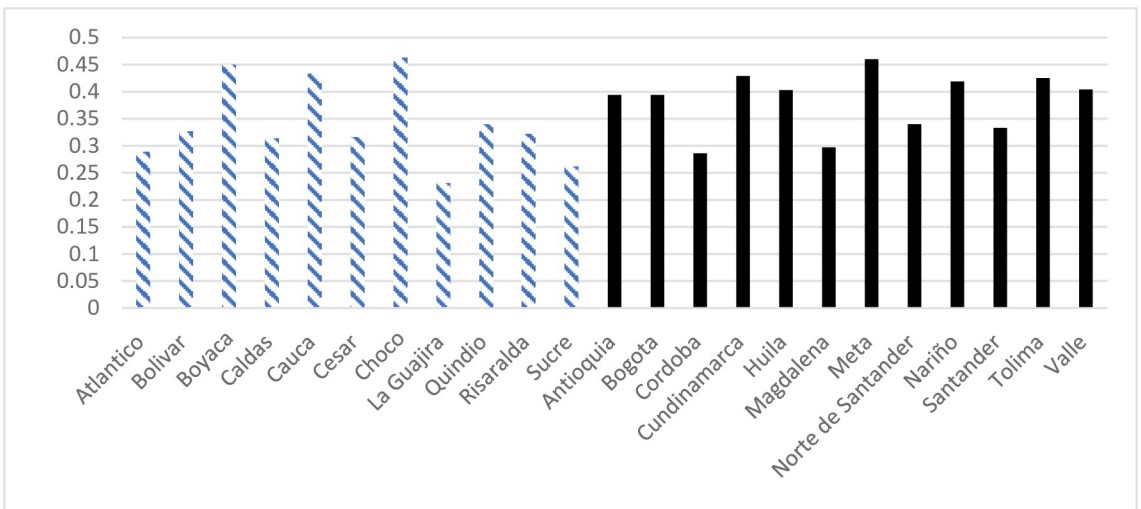

**Fig 2. Prevalence of self-reported IPV ever in 2010.** Treated departments in black untreated departments in stripes. Note: Weights for domestic violence used.

groups, but there are large differences within each group (Fig A4 in S2 File). By 2015, the index had declined to zero for all departments. Otherwise, there are only small but systematic differences between the treated and untreated departments. Women in the former group are better educated, wealthier, more likely to live in an urban area, and belong to non-minority groups (whites, mestizos, and castizos).

The last column reports standardized differences (STDIFFs), which indicate whether the covariates are able to adjust for differences in the two samples [37]. All the STDIFFS are relatively small, except for ln GDP per capita, which has a STDIFF of -0.87. This is on the high side, but this is partly due to the absence of sample weights, which cannot be used when calculating the normalized differences. Removing the two wealthiest departments, Bogota D.C. and Meta, reduces the STDIFF to 0.53, without affecting the findings (as Table 6 shows).

Table 2 provides the descriptive statistics of the departments. The second column shows the number of observations per department; it is important for inference that the sizes of the departments (clusters) are not too different, which is not the case. The ratio between the smallest and largest departments was 0.43.

The third and fourth columns show the GDP per capita in 2010 (in constant 2015 prices) and the percentage change between 2010 and 2015. As mentioned, the treated departments have, on average, higher GDP per capita, but there is a large variation in both groups. The growth rate between 2010 and 2015 also varies between departments but it was higher for the untreated departments (19.7%) than for the treated departments (16.8%).

The fifth column reports the Gini coefficients, which measure the degree of economic inequality. The differences between the groups are small, as are the differences in the changes in economic inequality between 2009 and 2015 (sixth column). The last two columns show a public sector management efficiency index reported by the UN agency CEPAL [41]. It ranks departments on a scale ranging from 1 to 100. Treated departments had a more efficient public sector in 2009, with a mean of 76 compared to 62. Note, however, that there were substantial changes between 2009 and 2015, which casts some doubt about the measure. Nevertheless, the difference in 2009 is probably partly a reflection of the presence of a gender policy, i.e., a better-functioning public sector is more likely to adopt new policies.

**Table 1. Mean values of control variables and standardized differences.**

| Variables | Treated | | Untreated | | STDIFF |
|---|---|---|---|---|---|
| | 2010 | 2015 | 2010 | 2105 | 2010 |
| Age, mean, 15–49 years | 34.56 | 34.36 | 34.04 | 33.64 | -0.02 |
| Education | | | | | 0.11 |
| No education | 0.02 | 0.01 | 0.03 | 0.03 | |
| Incomplete primary | 0.13 | 0.09 | 0.15 | 0.11 | |
| Complete primary | 0.16 | 0.11 | 0.14 | 0.11 | |
| Incomplete secondary | 0.22 | 0.19 | 0.23 | 0.21 | |
| Complete secondary | 0.26 | 0.27 | 0.26 | 0.24 | |
| Higher | 0.21 | 0.32 | 0.18 | 0.31 | |
| Wealth | | | | | 0.30 |
| Poorest | 0.15 | 0.15 | 0.25 | 0.27 | |
| Poorer | 0.19 | 0.19 | 0.24 | 0.25 | |
| Middle | 0.21 | 0.22 | 0.23 | 0.22 | |
| Richer | 0.22 | 0.22 | 0.18 | 0.16 | |
| Richest | 0.22 | 0.22 | 0.10 | 0.10 | |
| Ethnic group | | | | | 023 |
| None below | 0.88 | 0.88 | 0.78 | 0.80 | |
| Afro-descent | 0.09 | 0.08 | 0.16 | 0.11 | |
| Indigenous | 0.03 | 0.04 | 0.06 | 0.08 | |
| Small minority groups | 0.00 | 0.00 | 0.00 | 0.00 | |
| Father beat mother | 0.39 | 0.38 | 0.32 | 0.31 | -0.11 |
| Household size | 4.61 | 4.35 | 4.96 | 4.76 | 0.10 |
| Urban | 0.78 | 0.81 | 0.73 | 0.73 | -0.11 |
| Armed conflict | 1.73 | 0.00 | 1.71 | 0.00 | -0.25 |
| Ln GDP/capita | 16.48 | 16.64 | 16.15 | 16.31 | -0.87 |
| Observations | 23,423 | 16,958 | 8,885 | 6,611 | |

Note: Weights for domestic violence are used. The sources of GDP per capita and the index of armed conflict are [28, 38]. See Yang and Dalton [39] for a definition of standardized differences.

To conclude, most differences between treated and untreated departments and changes between 2010 and 2015 are likely to be captured by department fixed effects, individual and household control variables, GDP per capita, and the armed conflict index. I do not include the public sector efficiency index since it partly measures the presence of gender policy.

## Regression analysis

Table 3 reports the DiD estimates for samples of all women who have had a partner (S3 File reports the complete results). The upper panel shows the results for IPV in the past year, and the lower panel shows the results for IPV ever. Bootstrap p-values are reported in brackets. The estimates of the Post × Policy coefficients show a significant and negative effect in all regressions. When using data from the 23 preferred departments, the coefficient is -0.048 for IPV in the past year and -0.055 for IPV ever. Adding control variables only marginally changes the estimates. The results for the 31 departments are similar. Thus, the prevalence of IPV in the past year declined by close to 25% and IPV ever by about 15%.

Table 3 also shows that *Post* is insignificant and has a value close to zero in the regressions with IPV in the past year, which indicates that the decline in IPV was concentrated in departments with a gender policy. In the regressions with IPV ever, *Post* is positive and (borderline)

**Table 2. Descriptive statistics of departments.**

| Treated | Obs. | GDP/cap 2010 | % Change 2010–15 | Gini coeff. 2010 | %-change 2010–15 | Management index 2009 | %-change 2009–15 |
|---|---|---|---|---|---|---|---|
| Antioquia | 3,747 | 15.41 | 22.1 | 0.56 | -7.1 | 91.8 | -8.4 |
| Bogotá D.C. | 3,631 | 23.17 | 22.5 | 0.53 | -5.3 | 100 | -26.9 |
| Córdoba | 1,747 | 6.48 | 22.2 | 0.55 | -15.5 | 57.3 | 9.8 |
| Cundinamarca | 1,733 | 17.46 | 8.2 | 0.46 | -4.8 | 100.0 | -26.9 |
| Huila | 1,476 | 11.23 | 15.8 | 0.57 | -6.1 | 63.9 | 15.3 |
| Magdalena | 1,696 | 7.20 | 15.1 | 0.54 | -12.3 | 39.1 | 33.5 |
| Meta | 1,606 | 31.00 | 0.4 | 0.50 | -6.4 | 68.8 | 12.4 |
| Nariño | 1,611 | 5.78 | 31.6 | 0.50 | 0.2 | 78.8 | -14.7 |
| N. de Santander | 1,927 | 7.50 | 18.5 | 0.49 | -4.1 | 71.8 | -7.1 |
| Santander | 2,053 | 19.29 | 28.6 | 0.51 | -7.1 | 90.1 | -12.8 |
| Tolima | 1,552 | 10.72 | 22.7 | 0.55 | -8.0 | 80.0 | -10.4 |
| Valle del Cauca | 3,916 | 14.75 | 20.4 | 0.52 | -7.9 | 74.0 | -3.9 |
| Mean treated | | 14.17 | 16.8 | 0.52 | -7.0 | 76.3 | -3.3 |
| **Untreated** | | | | | | | |
| Atlántico | 2,243 | 11.16 | 33.8 | 0.50 | -11.3 | 72.8 | -9.5 |
| Bolívar | 1,914 | 11.32 | 24.5 | 0.51 | -4.9 | 48.2 | 16.4 |
| Boyacá | 1,548 | 14.62 | 27.1 | 0.54 | -0.4 | 93.0 | -17.5 |
| Caldas | 1,648 | 10.55 | 20.5 | 0.54 | -4.3 | 70.3 | 10.8 |
| Cauca | 1,625 | 7.40 | 39.2 | 0.57 | -7.3 | 61.3 | 6.2 |
| Cesar | 1,601 | 12.48 | 4.8 | 0.52 | -7.5 | 68.0 | -9.4 |
| Chocó | 1,659 | 7.43 | -5.7 | 0.57 | 4.7 | 19.8 | 62.6 |
| La Guajira | 1,519 | 11.28 | -4.3 | 0.61 | -10.1 | 35.3 | 55.0 |
| Quindío | 1,987 | 9.74 | 24.5 | 0.54 | -8.7 | 80.6 | 0.6 |
| Risaralda | 1,812 | 10.97 | 24.9 | 0.48 | -4.8 | 79.7 | -5.6 |
| Sucre | 1,905 | 5.71 | 32.7 | 0.54 | -12.3 | 56.4 | 1.4 |
| Mean untreated | | 10.24 | 19.7 | 0.54 | -6.1 | 62.3 | 10.1 |

Notes: The source of the GDP and Gini coefficients is DANE, the Office for National Statistics of Colombia [38, 40]. The index measuring public-sector institutional quality is based on several indicators related to fiscal performance and public-sector management of the governments of departments and major cities. It is published by CEPAL [41, 42].

significant when control variables are included, which indicates that there might have been an increase in IPV ever in departments without gender policy.

## Indirect evidence

Earlier research has uncovered a strong correlation between attitudes toward IPV and the prevalence of IPV [43]. Thus, one would expect a decline in positive attitudes toward IPV in departments with a gender policy. In the DHSs, women were asked the following questions: Is it acceptable that the husband beats his wife if she goes out without telling the husband, neglects her children, argues with her husband, refuses to have sex, and/or burns food? A binary variable measuring a positive response to any of these questions is used to estimate the effect of policy on attitudes.

Table 4 shows that *the Post x Policy* coefficient is -0.016, but not significant. This result is surprising, but it could be due to the low mean level in 2010, where only 3% had a positive attitude towards IPV; this can be compared to the fact that about 20% of the women reported being exposed to IPV in the past year. The weak effect could also be attributed to the

**Table 3. Effects of gender policy on intimate partner violence during the past year and ever, all women currently and formerly in a union.** DHS 2010 and DHS 2015.

| | 23 departments | | 31 departments | |
| --- | --- | --- | --- | --- |
| | IPV past year | IPV past year | IPV past year | IPV past year |
| Post x Policy | -0.048*** | -0.050*** | -0.049*** | -0.047*** |
| | (0.011) | (0.010) | (0.010) | (0.010) |
| Bootstrap p-value | [0.031] | [0.012] | [0.020] | [.016] |
| Post | 0.013 | -0.028 | 0.012* | -0.019 |
| | (0.008) | (0.171) | (0.016) | (0.015) |
| Bootstrap p-value | [0.327] | [0.442] | [0.289] | [0.623] |
| Mean in 2010 | 0.20 | 0.20 | 0.21 | 0.21 |
| | 23 departments | | 31 departments | |
| | IPV ever | IPV ever | IPV ever | IPV ever |
| Post x Policy | -0.055*** | -0.058*** | -0.054*** | -0.055*** |
| | (0.013) | (0.012) | (0.012) | (0.011) |
| Bootstrap p-value | [0.021] | [0.003] | [0.009] | [0.005] |
| Post | -0.002 | 0.042** | -0.003 | 0.043** |
| | (0.009) | (0.021) | (0.009) | (0.018) |
| Bootstrap p-value | [0.922] | [0.110] | [0.829] | [0.086] |
| Mean in 2010 | 0.37 | 0.37 | 0.37 | 0.37 |
| Department fixed effects | Yes | Yes | Yes | Yes |
| Controls | No | Yes | No | Yes |
| Observations | 46,156 | 44,518 | 56,461 | 54,350 |

Notes: OLS estimates are reported for DHS 2010 and DHS 2015. Survey design and weights were accounted for in all the regressions. *Post* is a survey dummy for DHS 2015. The control variables are five-year age dummies, ethnicity, educational attainment, father-beat mother, household size, urban residence, household wealth, Real GDP per capita, and index of armed conflict. The following departments were not included among the 23: Amazonas, Arauca, Caquetá, Casanare, Guainía, Guaviare, Putumayo, San Andrés y Providencia, Vaupés, and Vichada. * $p<0.1$; ** $p<0.05$; *** $p<0.01$.

interpretation of the questions about attitudes, that is, if they refer to their own attitudes or beliefs about norms in society [44].

Improved gender policies are also expected to affect behavior. Table 4, column (2), indicates that in the sample of women who have a partner, about two percentage points more lived with their partner in 2015 than in 2010 in treated departments compared to the others, although the bootstrap p-value is high (0.14). For the share of women who considered divorce or separation during the last 12 months, the treatment effect was -2.2 percentage points and significant (column 3). And among the women considering divorce or separation, the treatment effect was -8 percentage points for the share who indicated IPV as the main reason, but there was also a large positive overall effect (column 4), as indicated by *Post*. Since there is no evidence of an increase in IPV, these results could be accounted for by the reduced tolerance of IPV, in general, combined with a reduction in IPV in departments with a gender policy. Finally, column (5) shows that there was a significant treatment effect of 3.2 percentage points in the share of women who requested help from formal institutions, such as police, family commissary and courts, but an overall decline. Unfortunately, I could not test if there were changes in requests for help from friends and relatives since there are only data for these factors in DHS 2015.

Next, I use the strong association between accusations of unfaithfulness and IPV to evaluate the impact of gender policy. Although accusations of unfaithfulness can be due to a variety of reasons, such as male dominance, suspicion of infidelity is likely to be common and a key

**Table 4. Differences in differences coefficients, other responses.**

| VARIABLES | (1) Attitudes | (2) Lives with partner | (3) Plans divorce/separation | (4) Plans divorce due to IPV | (5) Asked for help |
|---|---|---|---|---|---|
| Policy x Post | -0.016 | 0.012* | -0.022*** | -0.080*** | 0.032** |
| | (0.010) | (0.007) | (0.008) | (0.016) | (0.015) |
| Bootstrap p-value | [0.180] | [0.141] | [0.009] | [0.002] | [0.057] |
| Post | 0.033* | 0.005 | -0.002 | 0.251*** | -0.067** |
| | (0.016) | (0.032) | (0.012) | (0.039) | (0.026) |
| Bootstrap p-value | [0.137] | [0.960] | [0.732] | [0.000] | [0.006] |
| Mean in 2010 | 0.03 | 0.94 | 0.28 | 0.29 | 0.25 |
| Observations | 46,156 | 34,750 | 34,750 | 8,352 | 7,589 |

Notes: See Table 3

\* $p<0.1$

\*\* $p<0.05$

\*\*\* $p<0.01$

trigger for IPV [35]. Since gender policy is unlikely to significantly affect unfaithfulness or men's suspicion of unfaithfulness, reduced IPV should manifest as a change in the association between them.

Fig 3 displays the estimates for IPV in the past year using predictive margins. As evident, accusations of unfaithfulness and IPV are highly correlated, IPV past year was about 30

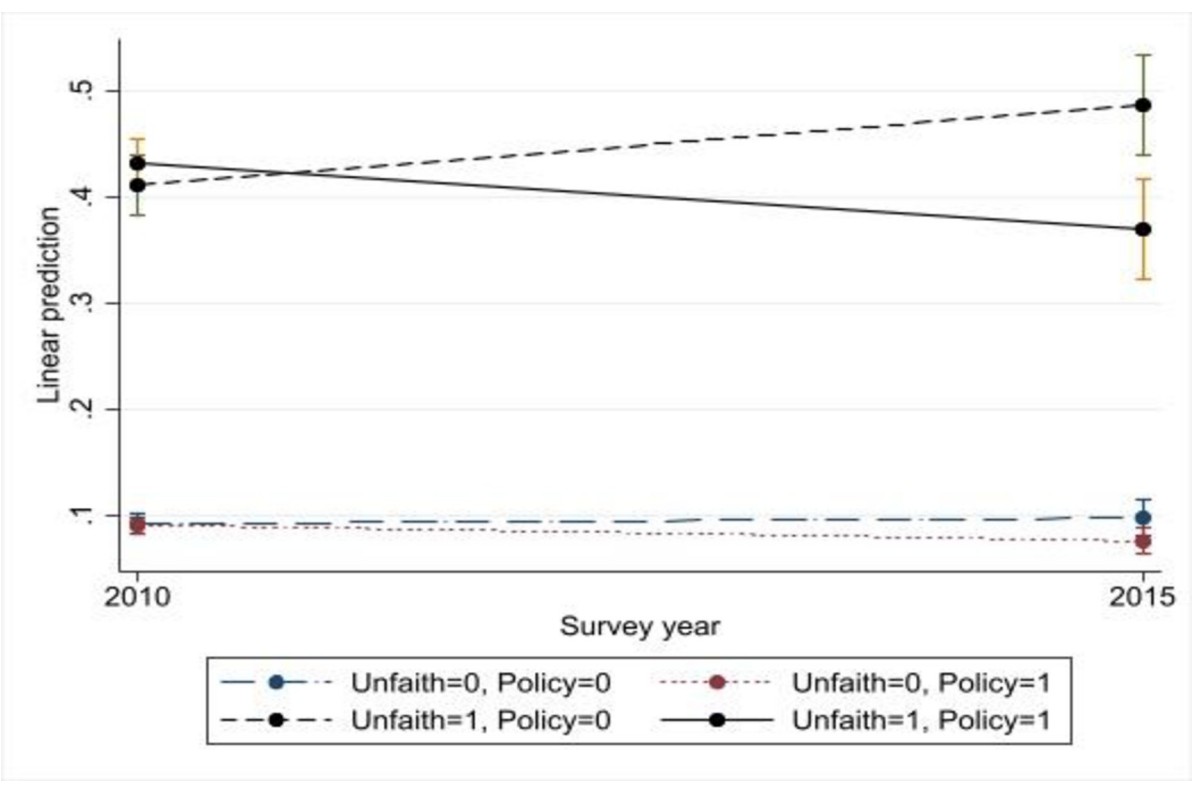

**Fig 3. Predictive margins for IPV past year and unfaithfulness based on the triple-differences model.** Note: Weights for domestic violence and robust standard errors clustered at the department level were used. See Table 3 for a list of control variables.

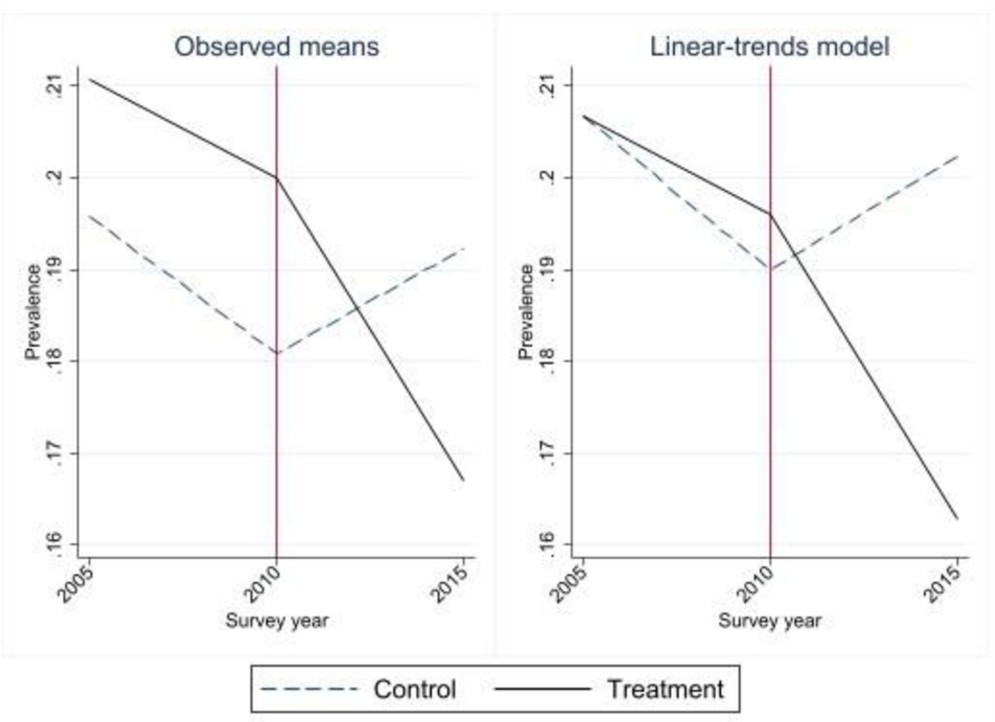

**Fig 4. Graphical diagnostics for parallel trends, 2005–2015, IPV past year, 23 departments.**

percentage points higher in 2010 among women accused of unfaithfulness than the others. Fig 3 also shows a decline in the association between unfaithfulness and IPV between 2010 and 2015 in treated departments, while there is an increase in the other departments. When there are no accusations of unfaithfulness, the predicted IPV declines somewhat in the treated departments, but there is no change in the others. The results for IPV ever are similar but somewhat weaker (see Fig A7 in S4 File).

## Assumptions and robustness checks

The estimation strategy based on Eq (1) relies on the assumption of parallel trends. In this section, I evaluate this assumption using data from DHS 2000 and DHS 2005, in addition to DHS 2010 and DHS 2015. Figs 4 and 5 plot the group-level trajectories of the means (left panel) and the results obtained with a linear-trends model that imposes a common reference point for the first survey year (right panel). The figures show that there is a faster decline in IPV in the past year for the non-treated groups between 2005 and 2010, while there are almost identical trends for IPV ever between 2000 and 2010. The results are similar for the sample with all 31 departments (Fig A8 in S5 File). Graphs of event study coefficients and their 95% confidence intervals, and F-tests for the pre-treatment period for the three models, provide further support for the parallel trends assumption: all the pre-treatment event study coefficients are close to zero and insignificant, and the null hypothesis of parallel trends is not rejected in any of the three tests (Figs A9 and A10, and Table A3 in S5 File).

The analysis of pretreatment trends is only indicative of the parallel trends assumption, which requires that the prevalence of IPV in the two groups of departments would have been on parallel trajectories in the post-treatment period without treatment. Furthermore, pretrends tests usually have low power [45]. Thus, I use the procedures developed by Rambutan

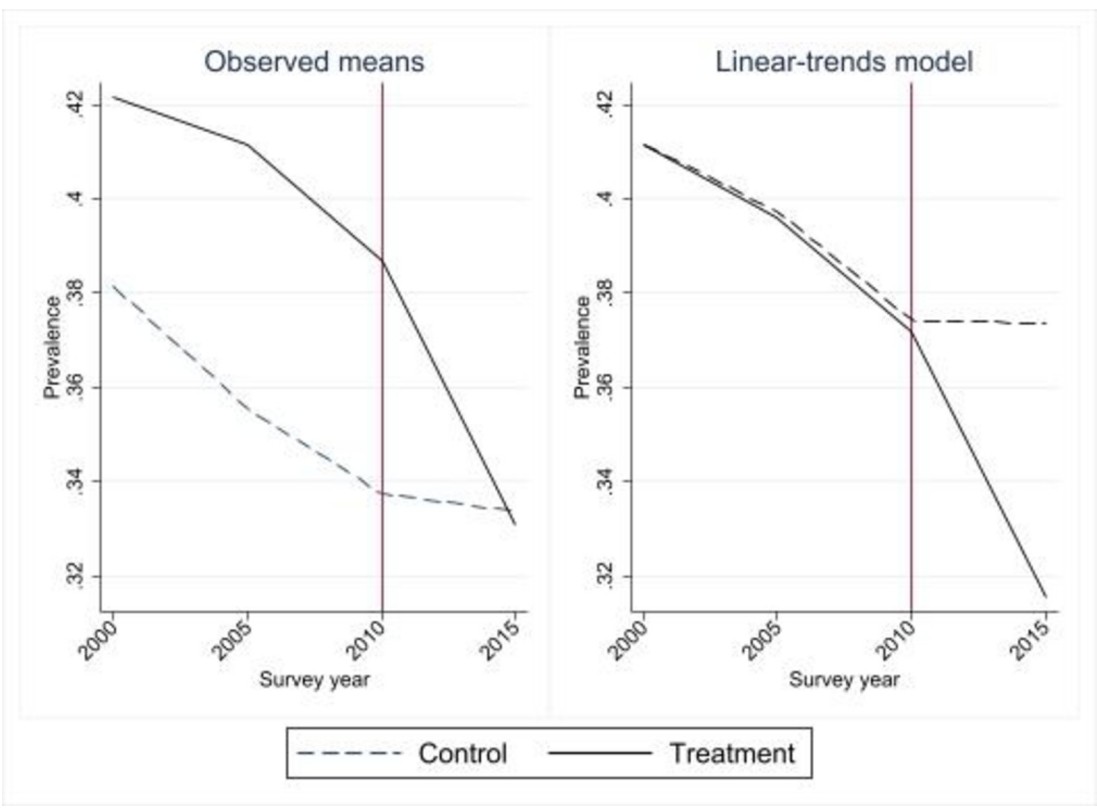

**Fig 5. Graphical diagnostics for parallel trends 2000–2015, IPV ever, 23 departments.**

and Roth [36] to evaluate the sensitivity of the treatment effects to two types of violations of the parallel trends assumption.

Fig 6 reports robust 95% confidence intervals of the DiD estimates for IPV in the past year when deviations of the parallel trends are assumed to be a multiple of the maximal deviation observed in the pre-treatment period. When Mbar is equal to zero, that is, Mbar = original, the confidence interval is obtained under the assumption of exactly parallel trends; when Mbar = 1, the deviation from parallel trends is assumed to be the same as the maximal deviation in the pre-treatment period; and when Mbar = 2, the deviation is assumed to be twice as large as in the pre-treatment period. Thus, the estimate of the treatment effect holds for violations of the parallel trends assumption in the post-treatment period that are twice as large as those observed during the pre-treatment period; the breakdown value for a significant effect is Mbar = 2.5.

The other violation is due to shocks in the post-treatment period that generate changes in linear trends. This creates a nonlinear deviation from the parallel trend, where M measures the percentage point change per period in the difference between the trends of the two groups. Fig 7 shows the results for IPV over the past year. M = 0 corresponds to the linear violation of parallel trends observed in the pre-treatment period, which is the same as Mbar = 1 in Fig 6. The breakdown value for a significant effect is M = 0.04, that is, four percentage points. A four-percentage point change per period is quite large. See, for example, Ang [46], Miller et al. [47], and Hausman et al. [48] for comparable results. The analysis of IPV ever provides similar results (Figs A11 and A12 in S5 File).

It is not straightforward to find convincing arguments for factors that might generate systematic differences in trends between treated and untreated departments during the post-

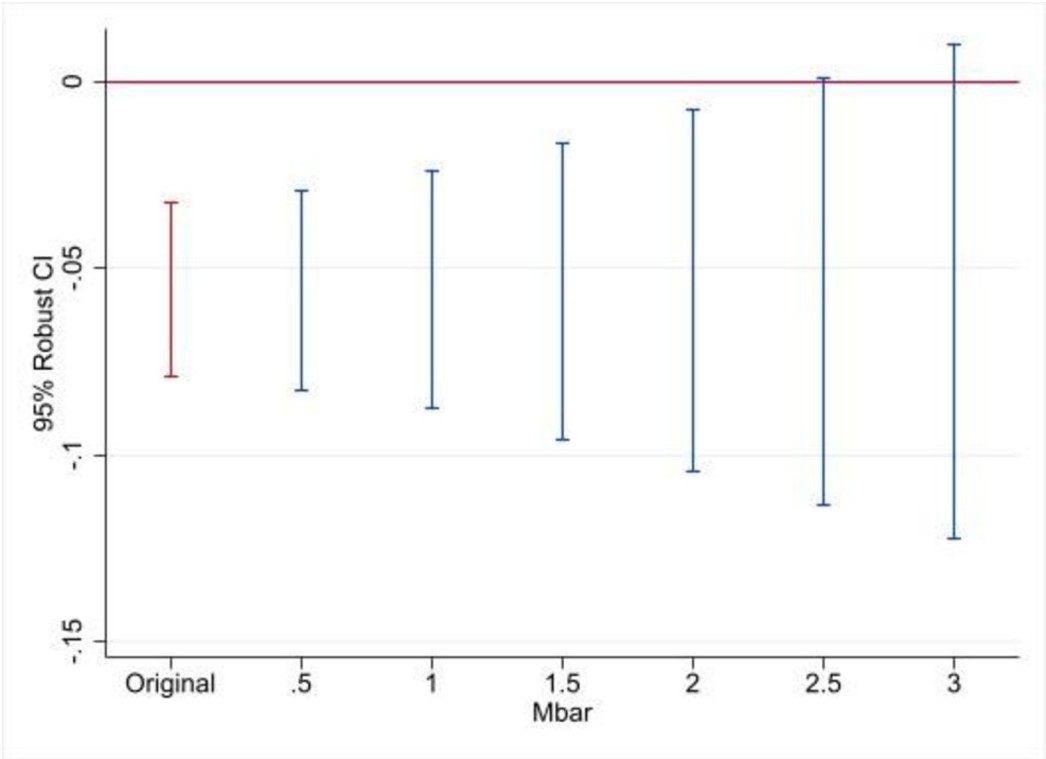

**Fig 6. Sensitivity analysis of IPV past year, violation of parallel trends in post-treatment period of Mbar times maximum deviation in pre-treatment period.** Note: The sample includes DHS 2005, DHS 2010, and DHS 2015. See Table 3 for a list of control variables.

treatment period. Two possible exceptions are armed conflicts and changes in income. Thus, the analysis above was conducted using an index of armed conflict and GDP per capita as control variables in addition to individual and household variables. Nevertheless, the same results are obtained without control variables (not reported). Thus, I conclude that the key findings in the study seem to be reasonably robust to deviations from the assumption of parallel trends, with and without control variables.

To evaluate the sensitivity of the specification of the gender policy variable, Table 5 reports estimates with narrow and broad specifications. The preferred specification, based on AECID [17], includes 12 of the 23 departments. The narrow specification includes only the eight departments that had a formal gender policy, that is, the four departments with only gender offices are excluded (Cordoba, Huila, Magdalena, and Norte de Santander), and the broad specification adds three departments that adopted gender policy programs between 2011 and 2013 (Atlántico, Bolivar, and Cauca). All regressions have a full set of control variables. The estimated treatment effects are the same with the narrow definition as with the original one, while they are about half the size with the broad definition and only significant when using robust standard errors clustered at the department level; both bootstrap p-values are greater than 010.

Next, I re-estimate the model for IPV in the past year, excluding individual departments. As Table 6 shows, there are only minor changes in the treatment effects, and all absolute t-values are higher than 3.4. The final row shows that excluding the two wealthiest departments, Bogota D.C. and Meta, does not affect the results.

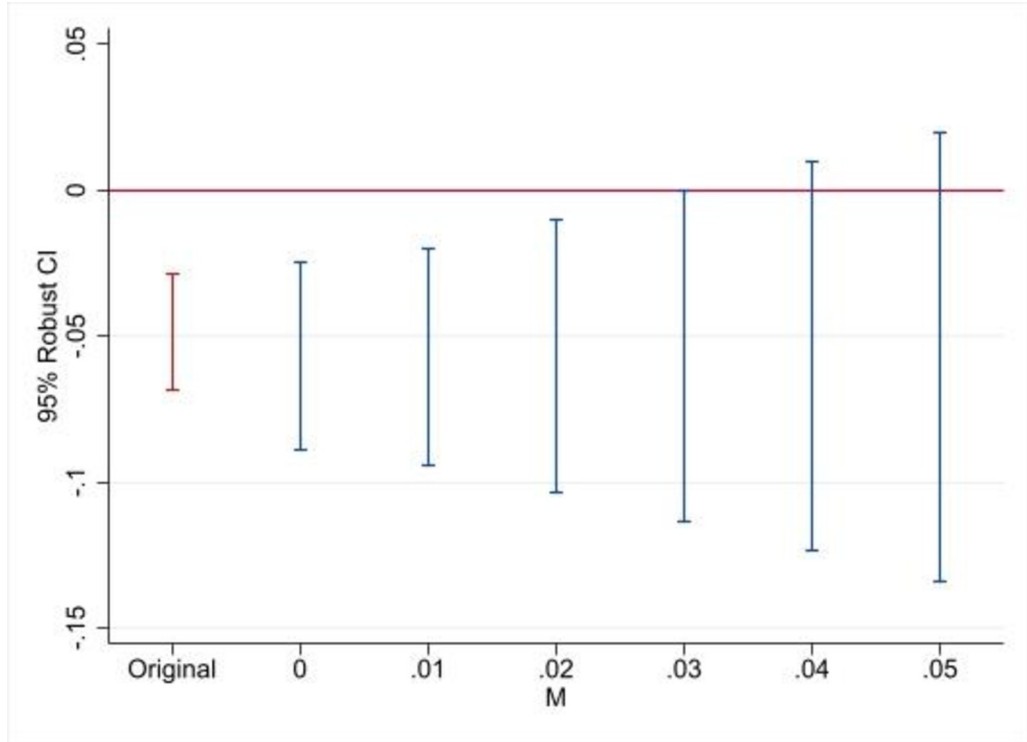

**Fig 7. Sensitivity analysis of IPV past year, violation of parallel trends in post-treatment period due to change in slope of M in each period.** Note: The sample includes DHS 2005, DHS 2010, and DHS 2015. See Table 3 for a list of control variables.

Finally, I report estimations with women in a union only (Table 7). The specification is the same as for previous models, except for the inclusion of age and education of the partner and the age difference between partners. All the treatment effects are significant but smaller than those obtained with women who have ever been in a union. The smaller impact of gender policies among women currently in a union than among women not in a union is what one would expect, since IPV is a common reason for divorce and separation.

## Discussion

This study provides estimates of the impact of gender policy in Colombia on physical and sexual violence between 2010 and 2015. Although women's advocacy groups were probably important drivers of the adoption of gender policy [49, 50], it was also closely linked to international agreements, such as the UN campaign "United to End Violence against Women, 2008–2015." Thus, roughly between 2008 and 2013, the government of Colombia strengthened the legal system by passing new laws and policies and by including IPV in national policy programs. A key component of this effort was to provide support to the governments of regional departments, which facilitates the adoption of departmental gender policies. To estimate the impact of the policy reforms, I exploit the fact that less than 30% of the departments had a gender policy in place in 2011.

The main finding of this study was that gender policies have a significant impact on IPV. In departments with a gender policy, physical/sexual violence during the past year decreased from 20% to 16% between 2010 and 2015, while it remained at 0.19 in the other departments. These estimates may be a lower bound in that gender policy departments' reporting of IPV

**Table 5. Alternative measures of treated departments; Narrow includes 8 departments and Broad 13 departments.**

|  | Narrow definition | | Broad definition | |
|---|---|---|---|---|
|  | **(1)** | **(2)** | **(3)** | **(4)** |
|  | IPV past year | IPV ever | IPV past year | IPV ever |
| Narrow x Post | -0.051*** | -0.050*** |  |  |
|  | (0.011) | (0.016) |  |  |
| Bootstrap p-value | [0.020] | [0.008] |  |  |
| Broad x Post |  |  | -0.027** | -0.023* |
|  |  |  | (0.011) | (0.013) |
| Bootstrap p-value |  |  | [0.152] | [0.186] |
| Post | -0.035* | 0.052** | 0.012 | 0.012 |
|  | (0.019) | (0.023) | (0.020) | (0.020) |
| Bootstrap p-value | [0.416] | [0.063] | [0.179] | [0.631] |
| Mean in 2010 | 0.21 | 0.38 | 0.20 | 0.37 |
| Observations | 37,952 | 37,952 | 44,518 | 44,518 |
| Depart. fixed eff. | Yes | Yes | Yes | Yes |
| Covariates | Yes | Yes | Yes | Yes |
| Number departs. | 18 | 18 | 23 | 23 |

Note: Narrow excludes four departments with only gender offices from the treated group (Cordoba, Huila, Magdalena, and Norte de Santander). Broad adds three departments that adopted gender policy programs between 2011 and 2013 to the treated group (Atlántico, Bolivar, and Cauca). For the estimates with Narrow, the control group is the same as in Table 3. See Table 3 for a list of control variables

*** p<0.01

** p<0.05

* p<0.1.

can become more socially acceptable, which would therefore increase the reported rates of IPV.

Additional analyses show that women who live in treated departments are less likely to have considered divorce or separation in the previous year. Among the women who had considered divorce or separation during the past year, there was a large increase in the share who

**Table 6. Removal of one department at the time, IPV past year.**

| Excluded department | *Post x Policy* | t-value | Excluded department | *Post x Policy* | t-value |
|---|---|---|---|---|---|
| Antioquia | -0.05 | -4.41 | La Guajira | -0.06 | -4.58 |
| Atlántico | -0.04 | -3.48 | Magdalena | -0.06 | -4.84 |
| Bogotá D.C. | -0.06 | -4.87 | Meta | -0.06 | -4.91 |
| Bolívar | -0.06 | -4.54 | Nariño | -0.06 | -5.04 |
| Boyacá | -0.07 | -5.43 | Norte de Santander | -0.06 | -5.07 |
| Caldas | -0.06 | -4.72 | Quindío | -0.06 | -4.99 |
| Cauca | -0.06 | -4.89 | Risaralda | -0.06 | -4.75 |
| Cesar | -0.06 | -5.10 | Santander | -0.06 | -4.99 |
| Córdoba | -0.06 | -4.68 | Sucre | -0.06 | -5.05 |
| Cundinamarca | -0.06 | -4.91 | Tolima | -0.06 | -4.72 |
| Chocó | -0.06 | -4.92 | Valle del Cauca | -0.06 | -4.71 |
| Huila | -0.06 | -4.86 | Bogotá D.C., Meta | -0.06 | -4.86 |

**Table 7. Differences in differences coefficients.** Women in a union at the time of the survey.

| | IPV past year | IPV past year | IPV ever | IPV ever |
|---|---|---|---|---|
| Narrow x Post | -0.044*** | -0.037** | -0.034*** | -0.027** |
| | (0.015) | (0.015) | (0.011) | (0.011) |
| Bootstrap p-value | [0.060] | [0.013] | [0.085] | [0.164] |
| Post | -0.013 | 0.001 | 0.000 | -0.005 |
| | (0.009) | (0.025) | (0.008) | (0.019) |
| Bootstrap p-value | [0.456] | [0.975] | [0.179] | [0.827] |
| Mean in 2010 | 0.16 | 0.16 | 0.31 | 0.31 |
| Observations | 30,379 | 29,369 | 30,379 | 29,369 |
| Depart. fixed eff. | Yes | Yes | Yes | Yes |
| Covariates | No | Yes | No | Yes |
| Number departs. | 23 | 23 | 23 | 23 |

Notes: In addition to the control variables described in Table 3, the age and education of the partner and the age difference between partners are included

* $p < 0.1$

** $p < 0.05$

*** $p < 0.01$.

reported IPV as the main reason, but the share in treated departments increased significantly less than in untreated departments. This can be due to the combined effect of reduced tolerance of IPV in general and a relative reduction of IPV in departments with a gender policy. Yet another piece of evidence of a change in behavior due to these policies is the decline in self-reported IPV in response to accusations of unfaithfulness in treated departments. Since it is unlikely that gender policy affects men's suspicion of unfaithfulness, and there does not seem to be a change, the decline in IPV may be due to behavioral change.

This study contributes to research by evaluating the initiatives of the government of Colombia and the UN to reduce IPV. To the best of my knowledge, this is the only study addressing this issue quantitatively in Colombia or other parts of the world. A number of studies evaluate specific prevention programs in other countries, such as workshop activities, and some studies analyze other aspects of IPV in Colombia, such as the characteristics of the data on IPV [36, 37], whereas others focus on the consequences of IPV [38] and on armed conflict and IPV in Colombia [23, 24]. Yet, it is widely recognized that there are many interconnected drivers of IPV and that prevention requires comprehensive strategies [33]. A combination of laws and a national plan with a multi-sectoral program, which engages with a range of stakeholders, as in Colombia, is therefore considered the most promising approach to preventing gender-based violence [34, 35]

Because I evaluate a policy that includes several interventions, it is challenging to determine the relevant mechanisms at work. Moreover, the departments that had adopted a gender policy program early were on average wealthier and had more efficient public sectors than the others. Therefore, I cannot draw conclusions about the effects of gender policies in general with certainty. To successfully implement gender policies, both a relatively well-functioning government sector and resources are required. Nevertheless, this study shows that gender policy can have a significant and socially relevant effect on the prevalence of IPV. Since both the national programs of Colombia and the process of adopting departmental gender policy programs have continued after 2015, future studies will be able to evaluate their final effects as more data become available

## Supporting information

**S1 File. Treatment, and choice of treated and untreated departments.**
(DOCX)

**S2 File. Differences between treated and untreated departments.**
(DOCX)

**S3 File. Complete Table 3 Differences-in-differences regressions.**
(DOCX)

**S4 File. Unfaithfulness and IPV ever.**
(DOCX)

**S5 File. Assumption of parallel trends.**
(DOCX)

## Acknowledgments

I am grateful for helpful comments and suggestions by Ann-Sofie Isaksson, Joseph Vecci, Clara Villegas Palacio, Melissa Rubio-Ramos, two anonymous referees, and participants at the Development Economics seminar at University of Gothenburg and the Economic and Social Context of Domestic Violence conference at SITE, Stockholm.

## Author Contributions

**Conceptualization:** Dick Durevall.

**Data curation:** Dick Durevall.

**Formal analysis:** Dick Durevall.

**Funding acquisition:** Dick Durevall.

**Investigation:** Dick Durevall.

**Methodology:** Dick Durevall.

**Project administration:** Dick Durevall.

**Validation:** Dick Durevall.

**Visualization:** Dick Durevall.

**Writing – original draft:** Dick Durevall.

**Writing – review & editing:** Dick Durevall.

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
