## [Decision Letter · Decision Letter 0]

13 Jun 2023

PONE-D-23-05491Gender policy and intimate partner violence in ColombiaPLOS ONE

Dear Dr. Durevall,

Thank you for submitting your manuscript to PLOS ONE. After careful consideration, we feel that it has merit but does not fully meet PLOS ONE’s publication criteria as it currently stands. Therefore, we invite you to submit a revised version of the manuscript that addresses the points raised during the review process.

Major Revisions required

We look forward to receiving your revised manuscript.

Kind regards,

Verda Salman, PhD

Academic Editor

PLOS ONE

Additional Editor Comments:

Major Revisions are required

Reviewers' comments:

Reviewer's Responses to Questions

**Comments to the Author**

1. Is the manuscript technically sound, and do the data support the conclusions?

Reviewer #1: No

Reviewer #2: Yes

2. Has the statistical analysis been performed appropriately and rigorously? 

Reviewer #1: N/A

Reviewer #2: No

3. Have the authors made all data underlying the findings in their manuscript fully available?

Reviewer #1: Yes

Reviewer #2: Yes

4. Is the manuscript presented in an intelligible fashion and written in standard English?

Reviewer #1: No

Reviewer #2: Yes

5. Review Comments to the Author

Reviewer #1: The introduction actually looks like an expanded summary of the study. The most important thing is that it does not have an explicit objective.

Restructure the manuscript to the traditional structure, include an objective, and resubmit.

The topic seems interesting, but without an explicit objective it is impossible to review further.

Reviewer #2: This is a good piece of research however, I have the following comments/suggestions for further improvement of the draft.

1. Introduction: Remove the details pertaining to methods and findings.

2. Methods: Why fixed effects? Apply Hausman test to validate the decision. How the econometric method used handles the endogeneity- add explanation.

3. Empirical Model: You should also include variables like 'religion', 'gender inequality', and 'depression' to improve the specification.

4. Table 1: Why 'no eduction' is repeated?

5. Information on the econometric method used is missing (I am not talking about DiD).

6. Diagnostic statistics of the models are missing.

6. PLOS authors have the option to publish the peer review history of their article (what does this mean?). If published, this will include your full peer review and any attached files.

Reviewer #1: No

Reviewer #2: **Yes: **Tanweer Ul Islam

---

## [Author Response · Author response to Decision Letter 0]

29 Jun 2023

Reviewer 1

The introduction actually looks like an expanded summary of the study. The most important thing is that it does not have an explicit objective.

Restructure the manuscript to the traditional structure, include an objective, and resubmit.

I apologize for the confusion. The structure of papers varies across disciplines, and since PLOS ONE is a multidisciplinary journal, I wrote the Introduction in a way that is common in my discipline. However, the paper is about global health, so I agree that the Introduction, in particular, should be changed. Moreover, the Introduction was not in line with the guidelines provided by PLOS ONE. I have now rewritten the Introduction and added a Discussion that contains the old Conclusion and some new material. I think the structure of the rest of the paper is clear, it is based on the content of the paper, which is more extensive than what is usually the case in papers in natural and health sciences. 

Reviewer 2

1 Introduction: Remove the details pertaining to methods and findings.

I have improved the Introduction and removed the topics suggested.

2 Methods: Why fixed effects? Apply Hausman test to validate the decision. How the econometric method used handles the endogeneity- add explanation.

 The difference-in differences (DiD) approach requires a treated group and a control group. In the simplest DiD specification, these are measured by a variable that is 1 for the treated units and 0 for the others. This variable is then interacted with a variable that is 1 during the treatment period and 0 otherwise to obtain the DiD variable. I use a more flexible specification where I allow the provinces to have separate intercepts; these are the fixed effects (I mention this because the terminology varies across disciplines). The fixed effects capture all omitted variables that change little or not at all during the study period and eliminate all data pooling. The models I estimate are called two-way fixed effects models because I also have time (survey) fixed effects. These models are frequently used when applying the DiD approach. Nevertheless, the results differ only slightly between the two specifications when one has few time periods, as in my case. 

It is, of course, possible to use random effects instead of fixed effects, but then one risks getting biased estimates due to omitted variables. Moreover, random effects models partially pool the data, while the DiD approach focuses on changes within each unit over time (department in my case), and this is obtained by using fixed effects. 

The issue of endogeneity is primarily related to the assumption of parallel trends, that is, that the outcomes of the treated and untreated groups would have followed the same trend on average if the treated group had not been treated. Of course, this is not testable; however, I evaluate the assumption as carefully as possible and check the robustness of the results. 

The use of control variables primarily aims to remove possible (minor) differences between trends and convince the reader that the results are robust. Moreover, even when the parallel trends assumption holds without control variables, adding them can increase the precision of the estimates. 

3. Empirical Model: You should also include variables like 'religion', 'gender inequality', and 'depression' to improve the specification.

This comment relates to the choice of control variables. Unfortunately, I do not have any information on religion or depression. I assume that most Columbians belong to the Catholic fate and that they did not see it as a variable of interest; a DHS has many questions. Moreover, education, wealth, urban/rural, and ethnic group variables are likely to capture the differences across religious groups, if any.

There is a depression module in the Columbian DHSs, but in contrast to many other DHSs, it is only used for the elderly. 

I have been careful not to include variables that can be outcomes of the treatment, since they might lead to biased results. The programs evaluated aimed to reduce gender inequality, so gender inequality could have been a dependent variable. Indeed, a reduction in IPV is a reduction in gender inequality. 

4 Table 1: Why 'no eduction' is repeated?

This was a mistake that I have corrected. It should have been higher education.

5 Information on the econometric method used is missing (I am not talking about DiD).

I assume you are referring to the estimator. I use OLS or, in other words, estimate linear probability models (LPMs). I now make this clear page 14 where I write

“The linear probability model, i.e., OLS, is used to estimate the DiD regressions. The DHS sample design is accounted for by using clustering and weights, as recommended (48).”

Although one could estimate logistical models, the nonlinearity makes it challenging to interpret the effects of the treatment, so they are normally not used. In any case, logistical and LPMs usually yield similar results. 

6 Diagnostic statistics of the models are missing.

I agree that I do not perform any standard regression diagnostic tests. However, there is no point in testing for linearity, normality, and homoscedasticity, since the dependent variable is binary. Linearity is ”imposed,” as it were, and the residual is either 0 or 1. Since I use survey data, independence is dealt with using methods that adjust for the stratified samples, that is, by using clustering and weights, as recommended by the DHS. See page 14 and the reference to 

Croft TN, Marshall AM, Allen CK, Arnold F, Assaf S, Balian S. Guide to DHS statistics. Rockville: ICF. 2018;645.

I evaluate leverage in Table 6 and the other key DiD assumptions by estimating event study models (Figs A9-10), simulating the effects of deviations from parallel trends (Figs 6-7), and alternating the definitions of the treatment and control groups (Tables 5 and 7).

---

## [Decision Letter · Decision Letter 1]

17 Jul 2023

PONE-D-23-05491R1Gender policy and intimate partner violence in ColombiaPLOS ONE

Dear Dr. Durevall,

Thank you for submitting your manuscript to PLOS ONE. After careful consideration, we feel that it has merit but does not fully meet PLOS ONE’s publication criteria as it currently stands. Therefore, we invite you to submit a revised version of the manuscript that addresses the points raised during the review process. Please submit your revised manuscript by Aug 31 2023 11:59PM. If you will need more time than this to complete your revisions, please reply to this message or contact the journal office at plosone@plos.org. Please include the following items when submitting your revised manuscript:A rebuttal letter that responds to each point raised by the academic editor and reviewer(s). You should upload this letter as a separate file labeled 'Response to Reviewers'.A marked-up copy of your manuscript that highlights changes made to the original version. You should upload this as a separate file labeled 'Revised Manuscript with Track Changes'.An unmarked version of your revised paper without tracked changes. You should upload this as a separate file labeled 'Manuscript'.If applicable, we recommend that you deposit your laboratory protocols in protocols.io to enhance the reproducibility of your results. Protocols.io assigns your protocol its own identifier (DOI) so that it can be cited independently in the future. For instructions see: https://journals.plos.org/plosone/s/submission-guidelines#loc-laboratory-protocols. Additionally, PLOS ONE offers an option for publishing peer-reviewed Lab Protocol articles, which describe protocols hosted on protocols.io. Read more information on sharing protocols at https://plos.org/protocols?utm_medium=editorial-email&utm_source=authorletters&utm_campaign=protocols.

We look forward to receiving your revised manuscript.

Kind regards,

Verda Salman, PhD

Academic Editor

PLOS ONE

Journal Requirements:

Additional Editor Comments:

Minor Revision

Reviewers' comments:

Reviewer's Responses to Questions

**Comments to the Author**

1. If the authors have adequately addressed your comments raised in a previous round of review and you feel that this manuscript is now acceptable for publication, you may indicate that here to bypass the “Comments to the Author” section, enter your conflict of interest statement in the “Confidential to Editor” section, and submit your "Accept" recommendation.

Reviewer #1: (No Response)

Reviewer #2: All comments have been addressed

2. Is the manuscript technically sound, and do the data support the conclusions?

Reviewer #1: No

Reviewer #2: Yes

3. Has the statistical analysis been performed appropriately and rigorously? 

Reviewer #1: N/A

Reviewer #2: Yes

4. Have the authors made all data underlying the findings in their manuscript fully available?

Reviewer #1: Yes

Reviewer #2: Yes

5. Is the manuscript presented in an intelligible fashion and written in standard English?

Reviewer #1: No

Reviewer #2: Yes

6. Review Comments to the Author

Reviewer #1: The manuscript was not summarized and restructured as recommended in the first review round.

It contains 7 tables and 7 figures. It is not a thesis, it should have no more than 4 tables/figures. It was recommended to fit the manuscript to the traditional structure: Introduction, Methods, Results, Discussion, but it was not done. The author should have the ability to summarize the information he wants to show.

Reviewer #2: (No Response)

7. PLOS authors have the option to publish the peer review history of their article (what does this mean?). If published, this will include your full peer review and any attached files.

Reviewer #1: No

Reviewer #2: **Yes: **Tanweer Ul Islam

---

## [Author Response · Author response to Decision Letter 1]

20 Jul 2023

“The manuscript was not summarized and restructured as recommended in the first review round.

It contains 7 tables and 7 figures. It is not a thesis, it should have no more than 4 tables/figures. It was recommended to fit the manuscript to the traditional structure: Introduction, Methods, Results, Discussion, but it was not done. The author should have the ability to summarize the information he wants to show.”

I apologize for not understanding the width of the comments on the previous version of the paper; however, I did make substantial changes to the Introduction and Discussion/Conclusion. 

I have created a section called Materials and methods with two subsections: Data and outcomes and Method. In addition, I have moved the data descriptive section and two other sections to the Results section, which now contains four sub-sections: Data description and summary statistics, Regression analysis, Indirect evidence, and Assumptions and robustness checks. I hope that this is in line with the comment above.

The paper is about twice as long as articles published in medical journals such as the Lancet, but compared to many papers in social sciences and economics, it is short (30 pages with doubly spaced text). In fact, one reason why I submitted the paper to Plosone is that it does not restrict the number of pages, tables, or figures. Moreover, I included several of the tables and figures in response to an earlier Plosone referee report, and I do not think the paper will be strengthened by moving them to the online appendix.

---

## [Editor Report · Decision Letter 2]

7 Aug 2023

Gender policy and intimate partner violence in Colombia

PONE-D-23-05491R2

Dear Dr. Durevall,

We’re pleased to inform you that your manuscript has been judged scientifically suitable for publication and will be formally accepted for publication once it meets all outstanding technical requirements.

Kind regards,

Verda Salman, PhD

Academic Editor

PLOS ONE

Additional Editor Comments (optional):

Accepted for Publication
---

## [Editor Report · Acceptance letter]

10 Aug 2023

PONE-D-23-05491R2 

Gender policy and intimate partner violence in Colombia 

Dear Dr. Durevall:

I'm pleased to inform you that your manuscript has been deemed suitable for publication in PLOS ONE. Congratulations! Your manuscript is now with our production department. 

Kind regards, 

on behalf of

Dr. Verda Salman 

Academic Editor

PLOS ONE